# Effects of Tropomodulin 2 on Dendritic Spine Reorganization and Dynamics

**DOI:** 10.3390/biom13081237

**Published:** 2023-08-11

**Authors:** Balaganesh Kuruba, Nickolas Starks, Mary Rose Josten, Ori Naveh, Gary Wayman, Marina Mikhaylova, Alla S. Kostyukova

**Affiliations:** 1Voiland School of Chemical Engineering and Bioengineering, Washington State University, Pullman, WA 99164, USA; balaganesh.kuruba@wsu.edu (B.K.); nickolas.starks@wsu.edu (N.S.); ori.naveh@wsu.edu (O.N.); 2Program in Neuroscience, Department of Integrative Physiology and Neuroscience, Washington State University, Pullman, WA 99164, USA; mary.josten@wsu.edu (M.R.J.); waymang@wsu.edu (G.W.); 3Center for Molecular Neurobiology, ZMNH, University Medical Center Hamburg-Eppendorf, 20251 Hamburg, Germany; marina.mikhaylova@hu-berlin.de; 4AG Optobiology, Institute of Biology, Humboldt Universität zu Berlin, 10115 Berlin, Germany

**Keywords:** tropomodulin, tropomyosin, neuron, dendritic spines, actin, synaptic plasticity, spine morphology

## Abstract

Dendritic spines are actin-rich protrusions that receive a signal from the axon at the synapse. Remodeling of cytoskeletal actin is tightly connected to dendritic spine morphology-mediated synaptic plasticity of the neuron. Remodeling of cytoskeletal actin is required for the formation, development, maturation, and reorganization of dendritic spines. Actin filaments are highly dynamic structures with slow-growing/pointed and fast-growing/barbed ends. Very few studies have been conducted on the role of pointed-end binding proteins in the regulation of dendritic spine morphology. In this study, we evaluated the role played by tropomodulin 2 (Tmod2)—a brain-specific isoform, on the dendritic spine re-organization. Tmod2 regulates actin nucleation and polymerization by binding to the pointed end via actin and tropomyosin (Tpm) binding sites. We studied the effects of Tmod2 overexpression in primary hippocampal neurons on spine morphology using confocal microscopy and image analysis. Tmod2 overexpression decreased the spine number and increased spine length. Destroying Tpm-binding ability increased the number of shaft synapses and thin spine motility. Eliminating the actin-binding abilities of Tmod2 increased the number of mushroom spines. Tpm-mediated pointed-end binding decreased F-actin depolymerization, which may positively affect spine stabilization; the nucleation ability of Tmod2 appeared to increase shaft synapses.

## 1. Introduction

Neuronal communications occur at highly specialized membrane compartments between neurons known as synapses. Based on their effects, synapses are classified as either inhibitory (GABAergic) or excitatory (glutamatergic) synapses. During the early stages of neuronal development, the majority of excitatory synapses are found on the dendritic shaft and are regarded as excitatory shaft synapses [1]. As the synapses mature, recruitment of post-synaptic density (PSD) components enables the excitatory shaft synapses to develop into large stable spine structures (thin, mushroom, and stubby). PSD is a specialized type of membrane supported by scaffolding proteins [2]. A dendritic spine consists of a head with PSD and a neck connecting it to the dendritic shaft. Mushroom spines are considered to be mature, with large excitatory synapses and very little plasticity (no room for synaptic strengthening); conversely, thin spines have smaller excitatory synapses and greater plasticity (more room for synaptic strengthening) [3,4]. Stubby spines are considered to be immature [5].

Actin polymerization and depolymerization are critical for the shape and motility of many cell types. In neurons, the appearance/disappearance of actin-based membrane protrusions and their associated neural functions are controlled by multiple actin-binding proteins [2]. Actin remodeling is essential for the formation, development, and maturation of dendritic spines—actin-rich membranous protrusions arising from dendritic shafts which carry excitatory synaptic contacts, in turn, positively affecting synaptic structural plasticity [5,6,7]. Mushroom spines are characterized by large bulbous heads and narrow necks, whereas thin spines have smaller heads and thin necks [5]. Stubby spines lack a well-developed distinguishable head and have short/wider necks [5]. Spines are continuously changing shape in response to synaptic input; maintaining this flexibility requires a high actin turnover.

Neuronal development involves the formation of other specialized dendritic projections, such as filopodia and spinules. Filopodia may become progenitors for either dendritic branches or spines. Spinules are shorter than filopodia and reach out from the layer above PSD to glial cells. They are thought to be important for establishing the initial stages of interaction between dendrites and axons prior to the formation of a stable synaptic connection, thus playing a key role in synaptogenesis and dendritic growth [8,9].

Actin filaments (F-actin) exhibit dynamic properties of polarization with a slow-growing end (pointed end) and a fast-growing end (barbed end) [10]. The kinetics of actin polymerization are controlled by a multitude of actin-binding proteins. Depolymerization at the pointed end releases G-actin monomers to the growing barbed end and is a rate-limiting step for F-actin polymerization [11]. Proteins controlling pointed-end actin dynamics have been particularly scantily investigated with respect to their function in the nervous system. More specifically, only a few studies have been conducted on the role of the pointed-end binding proteins in spine formation and morphology changes [12,13,14,15].

Tropomodulin (Tmod) is an actin-binding protein with a central regulatory role in actin filament dynamics at the pointed end. Of the four known Tmod isoforms, tropomodulin 2 (Tmod2) is unique to the brain [16]. Tmod2 knockout studies in mice have been shown to cause hyperactivity, learning/ memory impairments and an increase in long-term potentiation [17]. Knockdown (KD) of Tmod2 led to an almost two-fold increase of the neurite length in N2a cells [12]; in primary neurons, Tmod2 KD resulted in the increase in spine/synaptic density and the decrease of the length and the branch number in dendritic arbor [15]. Overexpression of Tmod1 and Tmod2 affected dendritogenesis and spinogenesis in neurons in an isoform-dependent manner [14].

Tmods perform their regulatory functions by binding and capping the pointed end, thus blocking the pointed-end elongation and depolymerization in a tropomyosin-dependent manner [18]. Tropomyosin is a family of coiled-coil proteins that wrap around actin filaments in a head-to-tail fashion [19]. Tmod’s binding to the pointed end is mediated by two actin-binding sites, ABS1 and ABS2 [20,21], and two tropomyosin-binding sites, TMBS1 and TMBS2 [22]. In previous studies, we have shown that L29E and L134D mutations in Tmod2 (Figure 1) disrupted its ability to bind tropomyosin in both TMBS1 and TMBS2 [14,23]. It was also shown that residues 347–351 in Tmod2 (Figure 1) are necessary for actin binding in ABS2, and their removal led to a loss of binding in that site, which impaired the nucleating ability [23].

In this work, we studied the effects of Tmod2 overexpression in primary hippocampal neurons; we examined its effects on spine motility, reorganization of the excitatory spine and shaft synapses, and other actin-based structures, such as filopodia and spinules. We showed that wild-type (WT) Tmod2 overexpression led to a decrease in the number of thin, mushroom, and stubby spines but had no effect on the number of branched spines. Tmod2 overexpression resulted in an increase in the lengths of thin and mushroom spines. While there was no effect on the number of filopodia, the number of spinules increased. The overexpression of WT Tmod2 did not affect the number of excitatory shaft synapses. However, destroying the tropomyosin-binding ability of Tmod2 resulted in a drastic increase in excitatory shaft synapses when the mutated Tmod2 was overexpressed. Disruption of the actin-binding ability of Tmod2 resulted in an increased number of mushroom spines, pointing toward increased spine stabilization.

## 2. Materials and Methods

### 2.1. Analysis of Fixed Neurons Overexpressing Tmod

#### 2.1.1. Constructs

pCAGGs vectors for expression of Tmod2, WT, and the mutated Tmod2 ED and Tmod2 A2, with an N-terminal Clover fluorescent protein (ClFP), were characterized previously [14]. Plasmids were isolated using endo-free MidiPrep Kit from Qiagen (Hilden, Germany) and used for transfection of primary hippocampal neurons.

#### 2.1.2. Animals

Wistar rats Crl:WI (Han) (Charles River, Nordrhein-Westfalen, Germany) and Wistar Unilever HsdCpb:WU (Envigo, Nordrhein-Westfalen Germany) rats were used in this study. Sacrificing of pregnant rats (E18) for primary hippocampal cultures was conducted in accordance with the Animal Welfare Law of the Federal Republic of Germany (Tierschutzgesetz der Bundesrepublik Deutschland, TierSchG) and with the approval of local authorities of the city-state Hamburg (Behörde für Gesundheit und Verbraucherschutz, Fachbereich Veterinärwesen, from 21 April 2015) and the animal care committee of the University Medical Center Hamburg—Eppendorf.

#### 2.1.3. Primary Neuronal Culture Preparation and Transfections

Rat primary hippocampal cultures were prepared as described in [25] and plated in 12-well plates on poly-l-lysine (Sigma-Aldrich, #P2636, St. Louis, MO, USA)-coated glass coverslips. Cells were grown in an incubator at 37 °C, 5% CO_2_, and 95% humidity. Cultures were transfected on the 15th day in vitro (DIV15) with pCAGGs vector using lipofectamine 2000 (Thermo Fisher Scientific, Dreieich, Germany), the DNA/lipofectamine ratio was 1:2. The ratio of ClFP-pCAGGs/empty vector was 1:1. Before transfection, the original neuronal medium was removed. Neurons were transfected in BrainPhys medium supplemented with glutamine. Transfection mix was added for 1 h. After transfection, the medium was exchanged back to the original BrainPhys medium. As with previous findings [26], it has been observed that the transfection efficiency is low, typically ranging between 1 and 5%. 

#### 2.1.4. Immunocytochemistry

Cells on coverslips were fixed 24 h after transfection with 4% paraformaldehyde for 10 min, washed 3 × 10 min with phosphate-buffered saline (PBS), and permeabilized with 0.25% Triton X-100 in PBS for 10 min. Then coverslips were washed 2 × 5 min in PBS and blocked for 45 min with a blocking buffer (10% bovine serum albumin, 2% glycine, 0.2% gelatin, and 50 mM ammonium chloride in PBS). All procedures were conducted at room temperature (RT). Primary antibodies against a post-synaptic marker Homer1 (mouse clone 2G8, Synaptic Systems 160011, Göttingen, Germany) and a dendritic marker MAP2 (rabbit polyclonal antibodies 3254, Abcam, Waltham, MA, USA) were added in the blocking buffer, and cells were incubated overnight at 4 °C. Cells were washed 3 × 10 min before adding secondary antibodies (goat anti-mouse IgG Secondary Antibody Alexa Fluor 568 A-11031 and goat anti-rabbit IgG Secondary Antibody Alexa Fluor 647 A21245, Thermo Fisher Scientific, Waltham, MA, USA) in the blocking buffer and incubated for 1 h at RT. Finally, cells were washed 3–5 × 10 min in PBS, and coverslips were mounted on microscope slides with Mowiol, prepared as [25].

#### 2.1.5. Fluorescence Microscopy

Confocal microscopy of fixed primary cultures was performed at a Leica TCS SP5 confocal microscope (Leica Microsystems, Manheim, Germany). The microscope was controlled by Leica Application Suite Advanced Fluorescence software. Samples were imaged using a 63× oil objective (Leica, HC PL APO CS2 63×/1.40 oil). Fluorophores were excited with a 488 nm Argon laser, 561 nm diode-pumped solid-state laser, and a 633 nm He–Ne laser. Images were acquired at 1024 × 1024. For z-stacks, z-step size was set to 0.25 µm.

#### 2.1.6. Image Analysis

For each construct, 11–17 neurons (average number of traced dendritic fragments per cell was 4) from 2–3 independent hippocampal cultures were subjected to blind analysis. Dendrites and dendritic spines were traced using the NeuronJ plugin [27,28,29]. Spines, filopodia, spinules, and excitatory shaft synapses were counted and sorted manually. Confocal images obtained were analyzed using ImageJ—Fiji labs [30]. Spines and non-spine structures were distinguished by the presence of Homer1 as a post-synaptic marker. Spines were counted and measured based on the criteria mentioned previously [5]. A dendritic length of 40 µm was chosen across all the samples to normalize the data obtained from spine measurements.

Overexpression of Tmod2 was quantified by measuring of fluorescence intensity in dendrites (three dendrites per cell, three cells per transfection, three transfections) using ImageJ. There was no statistically significant difference in mean fluorescence intensity between neurons overexpressing WT Tmod2 and Tmod2 A2 (*p*-value 0.5721 by unpaired *t*-test). Fluorescence intensity had non-significant small decrease for neurons overexpressing Tmod2 ED (*p*-value 0.0352) (Figure A1).

### 2.2. Live Neurons Imaging and Analysis

#### 2.2.1. Constructs

Plasmids expressing RFP-WT Tmod2 and Tmod2 mutants (ED and A2) were constructed by moving the gene, which encodes Tmod2 (with and without the mutations), from ClFP-Tmod2 constructs to pCAGGS destination vectors containing N-terminal RFP, using two-step Gateway cloning protocol [31] provided in the manual (ThermoFisher Scientific, Waltham, MA, USA). For this, attb sites were added to the insert using primers:Tmod2_attb1 forward:5′-GGGGACAAGTTTGTACAAAAAAGCAGGCTTCATGGCGCTCCCCTTTCAAAAAGG-3′Tmod2_attb2 reverse:5′-GGGGACCACTTTGTACAAGAAAGCTGGGTCCTACCTCCTGTCTCCTTCAACTC-3′Tmod2 A2_attb2 reverse:5′-GGGGACCACTTTGTACAAGAAAGCTGGGTCTTAAACTCTCTTCTTTCGAACCAGG-3′

Sequence-verified constructs were transformed into NEB 5-alpha-competent *E. coli* (New England Biolabs, Ipswich, MA, USA) cells for amplification. To obtain transfection-grade plasmid DNA, transformed *E. coli* cells were grown in 250 mL culture flasks, and plasmids were isolated using PureLinkTM HiPure Plasmid Midiprep Kit (ThermoFisher Scientific, Waltham, MA, USA).

#### 2.2.2. Primary Neuronal Culture Preparation and Transfections

Animal experiments were conducted in compliance with Washington State University IACUC-approved protocols 03717-019 and 04409-006. Hippocampal neuronal cultures were prepared from equal numbers of P1 female and male Sprague-Dawley rat pups. Briefly, dissected hippocampi were collected in cold Hibernate A (Brain Bits). Then, the hippocampi were coarsely chopped and collected in the plating media (1% B27, 1% Glutamax, 10% horse serum, 2% HEPES, pH 7.5 in Neurobasal A medium) containing 0.25% Papain (Sigma #P3125) and 0.2% DNase (Sigma Aldrich #D5025) and incubated at 37° C for 20 to 25 min with mild shaking. After settling the tissue down, the media containing papain and DNase was removed, and warm plating media was added; then, the tissue was triturated 6 to 10 times with fire-polished glass pipette in this media to achieve suspension of single cells. The cells were counted and plated using plates previously coated with Poly-L-Lysine at a density 4.7 × 10^4^ cells/cm^2^ for 6-well plates, which were used for live imaging. The cells were counted and plated at the density of 3 × 10^4^ cells/cm^2^ for 24-well plates, used for live staining experiments, and 4.7 × 10^4^ cells/cm^2^ for 6-well plates, used for biochemistry experiments, on plates that were coated previously with Poly-L-Lysine. After 2 to 3 h of plating, the plating media was changed with growth medium (1% B27 and 1% Glutamax in Neurobasal A medium) and maintained at 37 °C and 5% CO_2_. On the day in vitro DIV4, the feeding media (1% B27, 1% Glutamax, and 5 μM cytosine-d-arabinofuranoside (AraC) in Neurobasal A medium) was added to neurons to constitute one-third of the total media. Cells were grown and maintained in a CO_2_ incubator at 37 °C, 5% CO_2_, and 95% humidity [32]. Cell cultures were transfected on DIV5-6 by combining Lipofectamine 3000 (ThermoFisher Scientific, Waltham, MA, USA) with the designed pCAGGS constructs (2 µg/well for six-well plates) and incubated for 30 min. Post-incubation, the media was replaced with the original feeding media. On DIV9, media in the cell cultures were replenished by aspirating 1 mL of media and replacing it with 1 mL of fresh growth media. Cells were imaged on DIV11.

#### 2.2.3. Fluorescence Microscopy and Imaging

Imaging of cell cultures for dendritic spine live image capture was performed using an Olympus IX81 inverted confocal microscope (Olympus optical, Tokyo, Japan) equipped with a Hamamatsu ORCA-ER-charge couple device camera, 60× oil-immersion lens with numerical aperture of 1.4 and resolution of 0.280 µm using Slidebook 5.5 Digital microscopy software. Dendritic spine z-stack images were processed using MetaMorph software from Molecular Devices (Sunnyvale, CA, USA). 20-min time-lapse across the Z-stacks with a step size of 0.4 µm and number of steps being 4, were recorded for dendritic fragments. 1 mM Ascorbic acid was added into the chamber to prevent photobleaching events.

#### 2.2.4. Analyzing Changes in Spine Motility

Twenty-minute time-lapses of measured dendrites from experimental plasmid overexpressed neurons obtained as metadata (.nd) from confocal microscope were analyzed using Fiji ImageJ [30]. The metadata files were Z-stacked into image-J at maximum intensity. Brightness and contrast of the image were adjusted accordingly to highlight discernable differences between the background and the dendritic spines. Bleach-correct plugin [33] using histogram-matching algorithm was used to correct for any observed photobleaching events during the time-lapse capture and prevent any loss of data. StackReg plugin [34] was implemented as needed to correct any observed drifting motion. In order to track the spine movement and motility, MtrackJ plugin [35] was used to track all individually identified spines on one dendrite per cell over the course of 20 min. For manual spine selection, clearly distinct spines were chosen based on the first 5–10 frames in the 20 min/61 frame time-lapse. Tracking was carried out by manually choosing the center of the spine head with a circular point of 10 pixels in size. Tracking criteria were set to ‘Snap feature’ of centroid identification within a 3 × 3-pixel range to focus on the center of the chosen spine. The center of spine head was carefully selected in a frame-by-frame basis, and the following parameters were extracted by measuring the tracked spines using the MtrackJ—distance to source, or original position (D2S), distance to previous position (D2P) and velocity (V), as a measure of the meandering index. To measure D2S, the total distance traveled by the spine head from its point of origin is calculated using the following equation:$$D2S=\overset{N-1}{\underset{i=1}{\sum }}d\left(\mathrm{Pi},\;\mathrm{Pi}+1\right)$$
where Pi = (Xi, Yi) represents the co-ordinates of the selected point in the frame. To measure the distance (D2P) between two frames for the same point—P(X1, Y1) and P(X2, Y2)—the following equation is used:D2P = sqrt [(X2 − X1)2 + (Y2 − Y1)2]
Instantaneous speed/velocity, in this case, is measured as follows,
V = d (Pi, Pi + 1)/Δt
where for a constant frame rate, Δt is considered the time interval between successive frames. The extracted parameters, D2S, D2P, and velocity, were obtained for each spine in dendrite chosen per cell.

For the analysis of motility, root mean square deviation (RMSD) for each spine’s D2S and D2P and root mean square of velocity (RMSV) were calculated. Further, the average and standard deviations of these parameters were determined. Then, 9, 7, 8, and 8 videos were analyzed for the control, WT Tmod2, Tmod2 A2, and Tmod2 ED, respectively. The total number of mushroom spines analyzed per video varied from 5 to 12 (control), 6–12 (WT), 6–12 (A2), and 4–12 (ED). The total number of thin spines analyzed per video varied from 8 to 15 (control), 7–16 (WT), 4–10 (A2), and 5–16 (ED).

#### 2.2.5. Analyzing Changes in a Shape of a Spine Head

To analyze changes in the shape of mushroom and thin spines, five videos per each of the four groups (control, WT Tmod2, Tmod2 ED, and Tmod2 A2) were analyzed using Fiji ImageJ software. Three mushrooms and three thin spines were selected in each t = 0 image based on image quality and likeness to ideal spine morphology. The horizontal (HD) and vertical (VD) diameters of the spine heads were measured in pixels using the Fiji line tool, and the ratio between them was calculated. These calculations were performed on the selected spines for images taken at time points 1 through 12, corresponding to the first 220 s of the live image recording.

#### 2.2.6. Analyzing Spinule Formation in Live Neurons

Primarily classified transverse spines originating on the main dendrite, which we defined as the thickest, longest, or most populous branch in the image. In time-lapses where multiple branches had roughly equal dimensions and populations, the branch with the most apparent mushrooms was analyzed. For time-lapses without any well-defined mushrooms on the main dendrite, the next best branch was analyzed. Spines originating on other branches were only considered for classification if their heads were close enough to interact with the main dendrite either directly or via spinule; 2 µm or less was deemed sufficiently close. The first 3–5 min (10–15 frames) were used to classify spines; mushroom populations after this period were considered fixed and were the only subjects of analysis, regardless of whether they disappeared or new mushrooms formed. Any spine within 2–3 µm of the edge of the frame during this period was excluded from the analysis. Small transverse spines with bases on the underside of dendrites were not considered for analysis as they could not reliably be classified; additionally, given their orientation and our depth of view, angled spinules from these structures would have a greater chance of being falsely rejected than an identical structure on the upper half of the dendrite. The ImageJ macro, ROI 1-click tools [36], was used to group mushroom spines and record the number and duration of their spinule events. Spine parameters, as described before [5], were used as the initial basis for separation of symmetrical mushrooms from non-mushroom spines. Mushrooms arising from branched spines were excluded from analysis. Large asymmetric spines with relatively stable heads bore a striking resemblance to mushrooms with one or more synapses, as shown in other studies [37]. Without synaptic markers, these structures could not be definitively classified. Given the proximity of axons to dendrites in hippocampal cultures, their tendency to form synapses with spines and dendrites, in conjunction with our still neuron images using synaptic markers, we considered these large structures to be potential mushrooms provided they met several criteria [38]. Large potential mushrooms needed to have a clear head and neck, could not undergo drastic morphology changes between frames, and could not have a split head for more than one-third of the time-lapse. All of these structures had faint thin branches–possibly axons—connected to relatively small head protrusions, which appeared related to the large potential mushroom’s head distortions; these connections had to be distinguishable and predictable so as to not be falsely identified as spinules. Large potential mushrooms with more than three of these connections, as well as those with one or more ambiguous connection, were considered to be unanalyzable and were excluded from the mushroom population.

##### Spine Selection for Spinule Analysis

For spinule analysis, we chose several criteria: (1) Spinules should have a minimum length of ~1 µm extending from the mushroom head. (2) Spinule intensity is close to the head that it originated from. (3) Spinules should not break or be absorbed by another structure during the event. A wide range of spinule morphologies were accepted, provided they did not become bigger than, or otherwise break, the head they originated from. Once all criteria were met, the spinule was marked with ROI 1-click tools [36] on every frame from its initial protrusion to complete retraction. For the number of spinules on the mushroom spines producing them, data were normalized to the total number of mushroom spines for each dendrite. Spinule duration was calculated as an average for each dendrite. Each group, consisting of the control, WT Tmod2, Tmod2 A2, and Tmod2 ED, was analyzed using four videos. The total number of mushroom spines analyzed per video varied from 3 to 14 (control), 3–9 (WT), 3–6 (A2), and 19–24 (ED).

### 2.3. Data Presentation and Statistical Analysis

Box and whisker plots (Box plots in SigmaPlot) were used to present data obtained for the control, WT Tmod2, and the mutants. The box and whisker plots use the five-number summary of a set of data. The summary consists of a minimum, first quartile, median, third quartile, and maximum.

Statistical analysis was conducted for the motility data using a comparative unpaired two-sample *t*-test analysis using GraphPad Prism (GraphPad Software, San Diego, CA, USA). Data were not normalized. Test was accepted for a parameter-based comparison of the control vs. WT Tmod2, control vs. Tmod2 ED, control vs. Tmod2 A2, WT Tmod2 vs. Tmod2 ED, WT Tmod2 vs. Tmod2 A2 and Tmod2 ED vs. Tmod2 A2 groups.

For the head shape analysis, no statistically significant effect of time on HD/VD ratio was observed. In all cases, *t*-test comparison of data from the first and last time points analyzed revealed *p* > 0.05, so data for t = 20 − t = 220 was collapsed across time into means of each group. Pairwise *t*-tests were run to compare the effect of each Tmod2 construct on mushroom and thin spine diameter ratios against the control. Pairwise *t*-tests were run to compare the effect of mutated Tmod2 on mushroom and thin spine diameter ratios against WT Tmod2.

For the spinule statistical analysis, a two-tailed unpaired *t*-test with equal variance was used to determine if there were significant differences in the number of spinules normalized by a number of total mushrooms (with and without spinules) between neurons expressing the control, WT Tmod2, and the mutants.

## 3. Results

### 3.1. Effect of Tmod2 Overexpression on Reorganization of Dendritic Spines

Previous studies on the effect of Tmod2 overexpression on dendritic spine formation were performed by transfecting the hippocampal neurons on DIV6 and fixing them on DIV12 [14]. Here, we investigated the effect of Tmod2 overexpression on spine reorganization within 24 h by transfecting neurons on DIV15 when spines have already been formed.

In these experiments, we calculated the average numbers of thin, mushroom, and stubby spines (Figure 2, Table A1). Dendritic spine types were distinguished by their morphology, as in [5]. MAP2 and Homer 1 were used as specific markers for dendrites and for post-synaptic densities, respectively.

Changes in dendritic spine reorganization were observed by comparing neurons overexpressing ClFP-WT Tmod2 or ClFP alone as a control. Overexpression of WT Tmod2 in neurons resulted in a statistically significant decrease in the number of stubby spines (Figure 2C, Table A2). The total average length of thin and mushroom spines increased due to WT Tmod2 overexpression (Figure 3, Table A1 and Table A2).

Some of the spines can also branch at their neck to have the same or different spine head morphologies as a mechanism to increase the synaptic strength of a dendrite [5]. In this case, more than one post-synaptic density cluster was observed (Figure 3A). As seen in Figure 4C and Table A1 and Table A2, WT Tmod2 overexpression did not cause any statistically significant change in the number of branched spines.

### 3.2. Effect of Tmod2 Overexpression on Other Dendritic Actin-Based Structures

In developing neurons, excitatory shaft synapses are frequently located on the dendritic shaft, and as they mature, synapses relocate to dendritic spines [1,39]. However, some glutamatergic synapses remain on the shaft. Similar to spine synapses, they are enriched in F-actin [25]. Therefore, next, we asked what the effect of Tmod2 overexpression localization of synapses was. The average number of excitatory shaft synapses was calculated in neurons overexpressing WT Tmod2. We did not find any difference in the number of excitatory shaft synapses between cells overexpressing Tmod2 and control cells (Figure 4, Table A1 and Table A2).

In the areas of dense and active synapses, cellular protrusions arise from dendrites or spines to enhance synaptic plasticity [9]. Protrusions arising from dendrites are generally regarded as filopodia [8]. Smaller and thinner protrusions emerging from spines are known as spinules [6,9].

Average numbers of filopodia and spinules were calculated for neurons overexpressing WT Tmod2 (Table A1). Representative images of filopodia and spinules are shown in Figure 3. While there was no change in the number of filopodia, WT Tmod2 overexpression caused a statistically significant increase in the number of spinules (Figure 4B, Table A1 and Table A2).

#### 3.2.1. Effect of Disrupting Tropomyosin-Binding Abilities of Tmod2 on Reorganization of Spines and Other Actin-Based Structures in Dendrites

To investigate the effects of disruption of tropomyosin-binding sites in Tmod2 on spine reorganization and numbers of filopodia, spinules, and excitatory shaft synapses, we overexpressed Tmod2 ED similar to WT Tmod2 experiments.

Neurons overexpressing Tmod2 ED had no statistically significant difference in the numbers of thin and mushroom spines when compared to WT Tmod2 (Figure 2, Table A1 and Table A2). However, there was a slightly higher number of stubby spines than in neurons overexpressing WT Tmod2 (Figure 2B, Table A1 and Table A2). In comparison with the control, Tmod2 ED overexpression resulted in a decreased number of thin, mushroom, and stubby spines (Figure 2B, Table A1 and Table A2). There was no statistically significant difference in the length of thin and mushroom spines when compared to WT Tmod2 overexpression (Figure 3B, Table A1 and Table A2).

There was no change in the number of branched spines, filopodia, and spinules. Tmod2 ED overexpression resulted in an increase in the number of spinules when compared to the control (Figure 4B, Table A1 and Table A2). However, while overexpression of WT Tmod2 had no effect on the number of excitatory shaft synapses, overexpression of Tmod2 ED resulted in a significant increase in their number when compared both to overexpression of WT Tmod2 and the control (Figure 4B, Table A1 and Table A2).

#### 3.2.2. Effect of the Disrupting C-Terminal Actin-Binding Site of Tmod2 on Reorganization of Spines and Other Actin-Based Structures in Dendrites

To investigate the effects of the C-terminal actin-binding site disruption in Tmod2 on spine reorganization and numbers of filopodia, spinules, and excitatory shaft synapses, we overexpressed Tmod2 A2 similar to Tmod2 ED, WT Tmod2 experiments.

Tmod2 A2 overexpressed neurons had no statistically significant difference in the numbers of thin and stubby spines when compared to WT Tmod2 (Figure 2, Table A1 and Table A2). There was no effect on the length of thin and mushroom spines when compared to WT Tmod2 overexpression (Figure 3B, Table A1 and Table A2). There was no change in the number of branched spines, filopodia, spinules, and excitatory shaft synapses (Figure 4C, Table A1 and Table A2). However, overexpression of Tmod2 A2 resulted in a significant increase in the number of mushroom spines when compared to overexpression of WT Tmod2 and the control (Figure 2C, Table A1 and Table A2).

### 3.3. Effect of Tmod2 Overexpression on Dendritic Spine Motility

Dendritic spines are highly motile structures, as observed previously in many studies [40,41], and obtaining Tmod2’s effects on dendritic spine reorganization from static images alone only paints half the picture. To address that concern, we decided to evaluate the effects of Tmod2 on spine motility from 20 min time-lapses obtained from primary hippocampal neurons. The analysis was conducted for mushroom and thin spines. As we could not always distinguish between thin spines and filopodia without staining PSD components in our live-cell experiments, analysis for thin spines may include filopodia as well.

To quantify observed spine motility in our experiments, we calculated three parameters—distance to source/origin (D2S), distance to previous (D2P), and velocity (V), using MtrackJ plugin in ImageJ to track spine heads over time (Figure 5, Appendix A). The parameter D2S was used to reflect on the overall displacement or the traveling distance of the spine head to its initial position. D2P, on the other hand, refers to the fluctuating behavior of the spine head. V, depending on the context, refers to the frequency of the fluctuations (D2P) or the total rate at which the spine head is displaced (D2S).

Root mean square (RMS) values for the determined motility parameters, D2S, D2P, and V, were calculated for the primary hippocampal neurons overexpressing Tmod2. Each dendrite had different spine types with varying motilities (Figure 6A). The determined RMS values show the collective degree of the motion that is exhibited in all the spine heads across a dendrite. The obtained RMS values for the overall dendrites for the overexpressed neurons are plotted as shown in Figure 6B. To show the diverse motile behavior displayed by various spines in a dendrite that leads to the determined RMS values, we obtained the average RMSD_D2S_ of thin spines for several neurons (Figure 6B, Table A3). The dendrite with the RMSD_D2S_ closest to the average RMS was chosen as a representative dendrite to show the distribution of spine motility in a sample dendrite, as shown in the plots in Figure 6A.

RMSD_D2S_ and RMSD_D2P_ of spines were not statistically different in neurons overexpressing WT Tmod2 compared to the control. The RMSV for mushroom spines increased, but there was no statistically significant change for thin spines. (Figure 6B, Table A3 and Table A4).

Compared to WT Tmod2 overexpression, Tmod2 ED overexpression showed no statistically significant difference in RMSD_D2S_, RMSD_D2P_, RMSV of mushroom spines. However, Tmod2 ED-overexpressing neurons exhibited a statistically significant increase in the motility of thin spines (RMSD_D2S_, RMSD_D2P_, and RMSV) in comparison to WT Tmod2-overexpressing neurons (Figure 6, Table A3 and Table A4).

Overexpression of Tmod2 ED resulted in an increase in RMSV for mushroom spines with respect to the control, suggesting an increased destabilization and migration of mushroom spines. For thin spines, Tmod2 ED overexpression significantly increased RMSD_D2S_, RMSD_D2P_, and RMSV, indicating highly motile behavior brought upon by the disruption of tropomyosin-binding abilities of Tmod.

With respect to WT Tmod2-overexpressing neurons, there was no statistically significant difference in RMSD_D2S_ for thin and mushroom spines in Tmod2 A2-overexpressing neurons. However, there is a statistically significant increase in RMSD_D2P_ and RMSV of mushroom and thin spines, indicating an increase in their motility (Figure 6, Table A3 and Table A4).

### 3.4. Effect of Tmod2 Overexpression on a Shape of a Spine Head

Changes in spine head dimensions correlate with structural synaptic plasticity. By calculating the ratio between the horizontal (HD) and vertical diameters (VD) of spine heads, we were able to compare spines regardless of their actual size. On images, we considered a spine head to be elliptical in shape and measured HD and VD, as shown in Figure 7. In the control group (overexpression of the ClFP), an average shape of a mushroom spine head was closest to circular. Overexpression of WT Tmod2 resulted in a drastic increase in the ratio, making heads of mushroom spines more horizontally elongated (Figure 7A,B). Disrupting either actin-binding or tropomyosin-binding abilities (overexpression of Tmod2 A2 and Tmod2 ED, respectively) appeared to alleviate this effect; however, it still did not result in the effects seen in the control group.

In the control group, an average shape of a thin spine head is vertically elongated, while overexpression of WT Tmod2 made it horizontally elongated (Figure 7C,D). The overexpression of Tmod2 ED had a similar effect on thin spine heads as WT Tmod2, but to a lesser degree. However, overexpression of Tmod2 A2 resulted in more vertically elongated heads compared to the other groups, including the control.

The variability in HD/VD was calculated as standard deviations from the averages over a 225 s period (Table A5). The control overexpression had the smallest variability, while Tmod2 ED overexpression had the highest. All Tmod2 constructs caused some increase in variability, but only Tmod2 ED overexpression had a large statistically significant increase in variability for both mushroom spines and thin spines. The *p*-values for this increase were 0.0025 and 0.031, respectively, when compared to the control.

### 3.5. Effect of Overexpression of Tmod2, WT and the Mutants on Spinule Formation and Lifespan

To study the effects of Tmod2-dependent F-actin regulation on spinule dynamics, we decided to evaluate the 20 min time-lapse videos for two spinule parameters—number of spinules and average lifespan of a spinule in primary hippocampal neurons overexpressing control/WT Tmod2/Tmod2 ED/Tmod2 A2.

Due to various types of protrusions, we chose several criteria for these protrusions to be considered for our analysis (see materials and methods). Only mushroom spines were chosen for spinule analysis. From the previous results on fixed-image analysis, we noticed significant variations in the number of mushroom spines amongst the overexpressed neurons. To address these differences in numbers, we normalized the number of spinules, or spinule formations, to the total number of mushroom spines on each dendrite used for analysis (Table A6). We chose the *p*-value threshold to be ≤0.1 for the data to be statistically significant (Table A7). Overexpression of WT Tmod2 in neurons had no statistically significant difference in the spinule formation and lifespan compared to the control (Figure 8, Table A6 and Table A7).

There was no statistically significant difference in the number and duration of spinule events between neurons overexpressing WT Tmod2 and Tmod2 ED. However, if we compare Tmod2 ED-overexpressing neurons to the control, they have more spinule formations with no statistically significant difference in duration (Figure 8, Table A6 and Table A7).

There was also no statistically significant difference in the number and duration of spinule events between neurons overexpressing WT Tmod2 and Tmod2 A2 (Figure 8, Table A6 and Table A7). When compared to the control, Tmod2 A2 overexpression in neurons had significantly more spinule events with no statistically significant difference in duration (Figure 8, Table A6 and Table A7).

## 4. Discussion

Due to the dynamic nature of synaptic input, dendritic spines can rapidly change between a spectrum of spine morphologies ranging from standard—thin, mushroom, and stubby—to specialized types—branched, spinules/spine head protrusions [5]. These activity-dependent massive turnovers occur during developmental stages or by learning and changes in sensory experiences due to their elimination by de novo spine formation [42,43]. The shape of a dendritic spine head is dependent on synaptic strength [5]. The relationship between the dynamic turnover of spines and the synaptic strength is referred to as structural synaptic plasticity [44].

Actin cytoskeleton undergoes constant assembly and disassembly [45]. The alteration of actin dynamics by actin-binding proteins provides precise regulation in the formation and behavior of pre- and post-synapses [46]. The modes of regulation of actin dynamics responsible for spines’ active behavior include actin sequestering [47], activation of actin monomers [48], nucleation [49,50], capping [23], and stabilizing [51] actin filaments.

For the formation of highly complex and diverse dendritic structures, it is essential for G-actin to be available in high concentrations [52]. The dissociation of actin monomers at the pointed end is used as an additional source of G-actin for barbed-end polymerization. So far, the role of pointed-end capping of actin filaments in dendritic spine dynamics has not been fully understood. The tropomodulin family of proteins (Tmod) represents the most important pointed-end binding proteins, with Tmod2 being exclusively expressed in neurons.

### 4.1. Effects of Tmod2 on Dendritic Spine Formation during Synaptic Development

It was shown that Tmod2 levels remain fairly constant in N2a neuroblastoma and PC12 cells [12]. Expression of Tmod2 in primary hippocampal neurons in rats increases during embryonic stage 18 and remains steady all the way through adulthood [15]. Overexpression of Tmod2 in primary hippocampal neurons led to increased dendritic complexity and increased numbers of thin, mushroom, and stubby spines during the developmental stages, DIV7-12 [14]. Our live-imaging data demonstrate that overexpression of WT Tmod2 during development stages did not seem to affect spine motility (Figure 6 and Table A3 and Table A4). However, WT Tmod2 overexpression resulted in horizontal elongation of mushroom and thin spine heads (Figure 7), most likely due to both formation of new actin filaments and the stabilization of existing ones.

In Tmod2 ED, the capping ability was disrupted, while the actin-nucleation ability was preserved. In Tmod2 A2, the actin-nucleation ability was disrupted while the capping ability was preserved. Eliminating the capping ability of Tmod2 previously was shown to have no effect on the number or the shape/length of the spines with respect to WT Tmod2 [14]. However, disrupting the capping ability of Tmod2 led to a drastic increase in the motility of thin spines (Figure 6 and Table A3 and Table A4) and excessive flattening of thin spine heads. Lack of capping ability implies the take-over of the role of actin nucleation by Tmod2 ED. The increase in thin spine motility and variability in the shape of spine heads suggest that Tmod2’s capping ability plays an important role in stabilizing and strengthening thin spines. The opposite effects of Tmod2 A2 overexpression on shapes of thin and mushroom spine heads (Figure 7) indicate that Tmod2’s capping ability affects mature and immature spines differently.

Eliminating actin-nucleating ability in Tmod2 previously was shown to result in a decreased number of mushroom spines when compared to the control and WT Tmod2 overexpression [13]. In our live-imaging studies, impairing Tmod2’s actin-nucleating abilities resulted in a significant increase in the fluctuating motility (D2P and V) for mushroom spines with no statistically significant spine head displacement (D2S). These effects could be explained by a disproportionate takeover of capping vs. nucleating abilities in Tmod2 A2.

Overall, these effects on the motility parameters further suggest that capping ability plays an important role in spine maturation and stabilization.

### 4.2. Effects of Tmod2 on Dendritic Spine Reorganization

In this study, we explored the morphology-modulating effects of Tmod2 in the nervous system during the stages of synaptic reorganization, DIV 15–16. In contrast to the increase in the number of spines during development [14], WT Tmod2 overexpression resulted in a statistically significant decrease in the numbers of thin, mushroom, and stubby spines during dendritic spine reorganization. Increased activity in actin nucleation and capping due to WT Tmod2 overexpression could have led to an increase in the average length of existing spines due to amplified F-actin regulation activity by Tmod2.

Eliminating Tmod2’s capping ability did not result in any statistically significant differences in either the number or the length of the spines. However, there was a drastic increase in the number and length of mushroom spines due to the disruption of actin-nucleation ability, indicating signs of synaptic strengthening and spine stabilization. The branched actin in the spine head [2] may be positively affected by the prevailing capping ability of Tmod2 A2. These capping abilities can further stabilize already-formed F-actin filaments in the existing spines.

### 4.3. Effects of Tmod2 on Other Actin-Based Non-Spine Synaptic Structures

Excitatory shaft synapses are known to form prior to the formation of spinular synapses, thereby completely skipping the filopodial state of formation [53]. Tmod2 was previously reported to populate predominantly along the periphery of the dendritic shaft and in the neck and base of the spine head [15]. Disruption of Tmod2’s capping ability resulted in a significant increase in excitatory shaft synapses, which correlates to the takeover of Tmod2’s actin nucleating ability of overexpressed Tmod2 ED. This increase in shaft synapses appears to be a redistribution or retraction of actin from spines into dendritic shafts.

Dendritic spinules are thin, short-lived projections arousing from spine heads that are induced by synaptic activity to interact with presynaptic termini or other spines for communication or exchange of material [9]. Many actin regulation pathways are involved in spinule dynamics and development [37]. Though Tmod2 overexpression did not show any statistically significant differences in spinule events during the spine formation (Figure 8), there was a significant increase in the number of spinule events formed during synaptic reorganization (Figure 4, Table A1 and Table A2). While disrupting capping ability also has an increase in spinule event numbers, eliminating Tmod2’s nucleating ability had no effect. Tmod2’s capping ability appears to contribute towards the formation of spinule events only during developmental stages. There was no difference in the number of spinule events during synaptic reorganization (Figure 4 and Figure A2, and Table A1 and Table A2). Further studies are required to establish the role of Tmod2 in spinule-mediated long-term potentiation during synaptic reorganization.

## 5. Conclusions

Tmod2 is a brain-specific pointed-end capping protein with the strongest actin-nucleating ability of all Tmod isoforms [50]. Tmod1, also found in neurons [12], has very weak nucleating ability. Our findings suggest that Tmod2’s nucleating and capping abilities have very specific effects on dendritic spine formation, reorganization, and motility. Understanding the role of physiological effects of F-actin pointed-end regulation shall provide many insights into actin dynamics not just in synapse biology but also in many of the critical functions that occur in neurons. It is worth noting that when Tmod2 was knocked out in mice, it resulted in Tmod1 up-regulation in brain cells [17]. However, no change was observed in Tmod1 expression when Tmod2 was knocked down by about 50 % [15]. In future studies, understanding studying how Tmod2 functions together with Tmod1 and other pointed-end regulators, such as Tmod1 and CAP2 [54], or cofilin that regulates F-actin depolymerization at the pointed end [55] will allow us to understand the pointed-end mediated regulation of spine plasticity. 

## Figures and Tables

**Figure 1 biomolecules-13-01237-f001:**
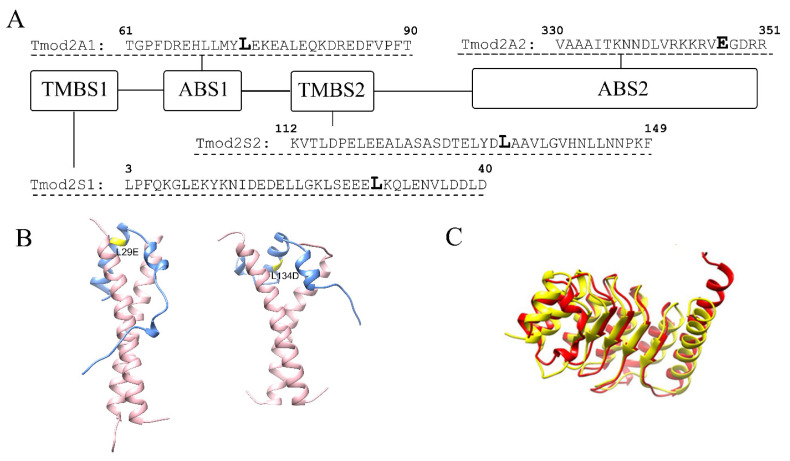
(**A**) A schematic presentation of Tmod2 binding sites. TMBS is a tropomyosin-binding site, and ABS is an actin-binding site. Bold L (Leu) residues in TMBS1 and TMBS2 indicate the location of Tmod2 ED mutations. Tmod2 A2, bold E (Glu) residue in ABS2 shows where a stop codon was introduced for truncated Tmod2 (Tmod2 A2). (**B**) Simulated models for Tmod2 ED showing the location of the L29E and L134D mutations (yellow) [24]. (**C**) Simulated models for Tmod2 WT (red) and Tmod2 A2 (yellow) showing the truncation. Reprinted with permission from [13]. 2018, Alla Kostyukova.

**Figure 2 biomolecules-13-01237-f002:**
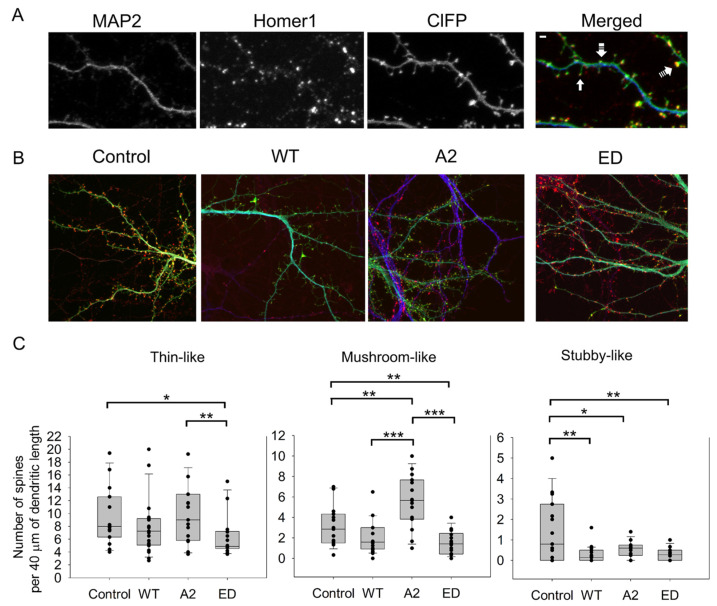
Effect of overexpression of WT Tmod2 and the mutants (Tmod2 ED and Tmod2 A2) on spine number. ClFP-tagged (green) Tmod2 was overexpressed in primary hippocampal neurons (transfected on DIV15 and fixed on DIV16). (**A**): Representative image of a dendrite of a neuron overexpressing ClFP (control) showing a dendritic marker MAP2 (blue) and a post-synaptic marker Homer1 (red). Block, double-dashed, and triple-dashed arrows indicate thin, stubby, and mushroom spines, respectively. Scale bar = 1 µm, shown in the merged image. (**B**): Representative merged images of neurons overexpressing ClFP, ClFP-Tmod2 WT, ClFP-Tmod2 ED, and ClFP-Tmod2 A2. (**C**): Average numbers of spine types per 40 µm of dendritic length. Error bars indicate the standard error of the mean (SEM). Asterisks indicate a statistically significant difference from the control, *, **, *** represents *p*-values of *p* ≤ 0.05, *p* ≤ 0.01 and *p* ≤ 0.001, respectively.

**Figure 3 biomolecules-13-01237-f003:**
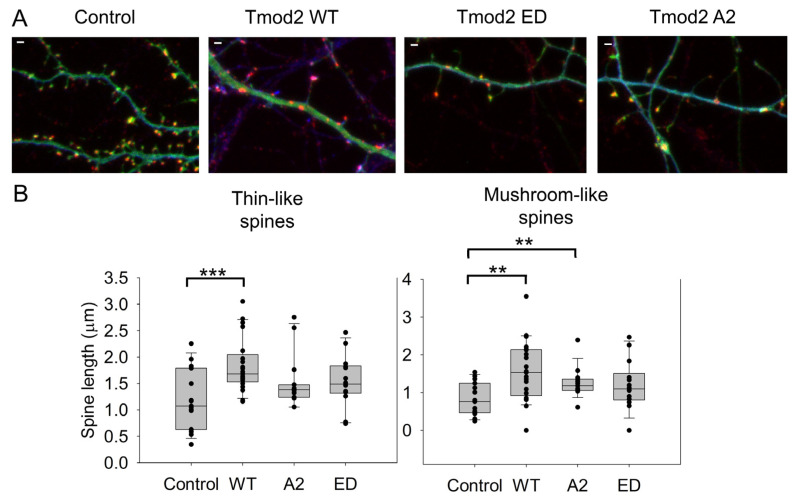
Effect of overexpression of WT Tmod2 and the mutants (Tmod2 EDTmod2 A2) on the average length of dendritic spines. ClFP-tagged Tmod2 (green) was expressed in primary hippocampal neurons (transfected on DIV15 and fixed on DIV16). (**A**): Representative images of dendrites of a neuron overexpressing ClFP (control), Tmod2 WT, Tmod2 ED, or Tmod2 A2. MAP2 (blue) is a dendritic marker, and Homer1 (red) is a post-synaptic marker. Scale bar = 1 µm. (**B**): Box plots for average lengths of thin and mushroom spines. Error bars indicate the standard error of the mean (SEM). Asterisks indicate statistically significant differences; ** and *** represent *p*-values of *p* ≤ 0.01 and *p* ≤ 0.001, respectively.

**Figure 4 biomolecules-13-01237-f004:**
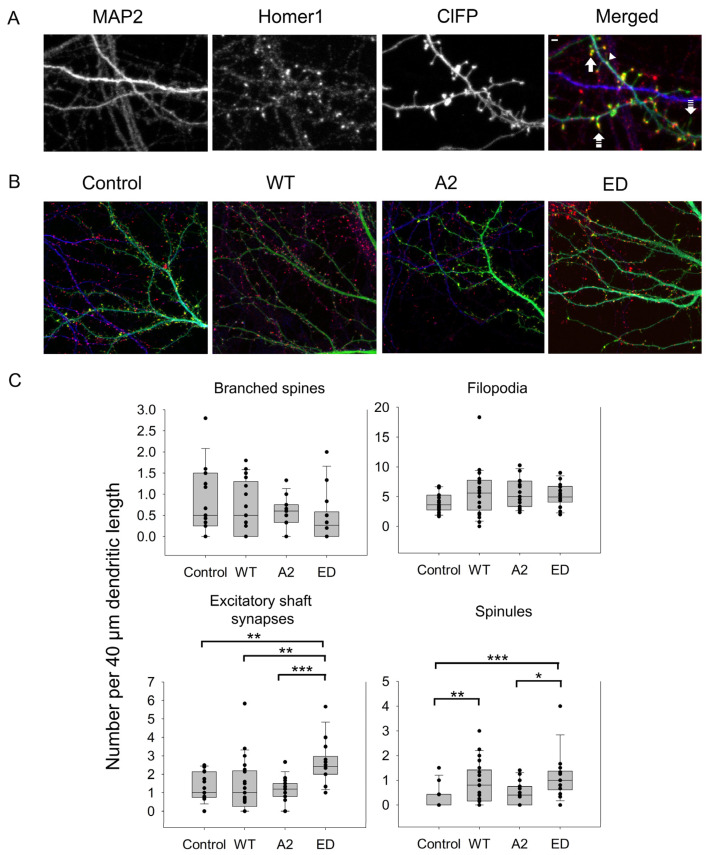
Effect of overexpression of WT Tmod2 and the mutants (Tmod2 ED and Tmod2 A2) on numbers of different actin-based dendritic structures. ClFP-tagged Tmod2 (green) was overexpressed in primary hippocampal neurons (transfected on DIV15 and fixed on DIV16). (**A**): Representative images of dendrites of a neuron overexpressing ClFP (control, upper panel) and Tmod2 WT (lower panel) showing a dendritic marker MAP2 (blue) and a post-synaptic marker Homer1 (red). Scale bar = 1 µm. Block, double dashed, triple dashed arrows, and arrowhead indicate branched, spinule, filopodia, and excitatory shaft synapses, respectively. (**B**): Representative merged images of neurons overexpressing ClFP, ClFP-Tmod2 WT, ClFP-Tmod2 ED, and ClFP-Tmod2 A2. (**C**): Box plots for an average number of branched spines, filopodia, excitatory shaft synapses, and spinules. Error bars indicate standard error of the mean (SEM). Asterisks indicate statistically significant differences; *, **, and *** represent *p*-values of *p* ≤ 0.05, *p* ≤ 0.01, and *p* ≤ 0.001, respectively.

**Figure 5 biomolecules-13-01237-f005:**
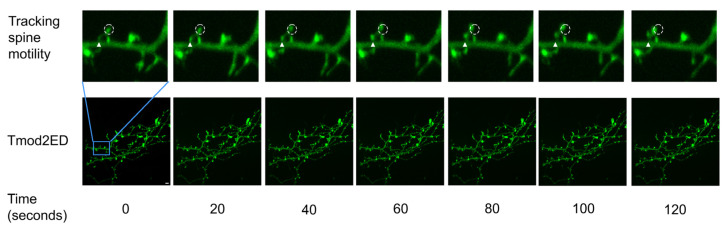
Representative 2 min section of a 20 min time-lapse (frames 1 to 7) for Tmod2 ED. The solid triangle shows the initial position of the spine base and the initial position for the tracked spine head is shown in the dashed circle, from which the motility parameters. Distance to source (D2S) is measured using MtrackJ plugin in ImageJ. Scale bar = 5 µm shown in Time 0 image.

**Figure 6 biomolecules-13-01237-f006:**
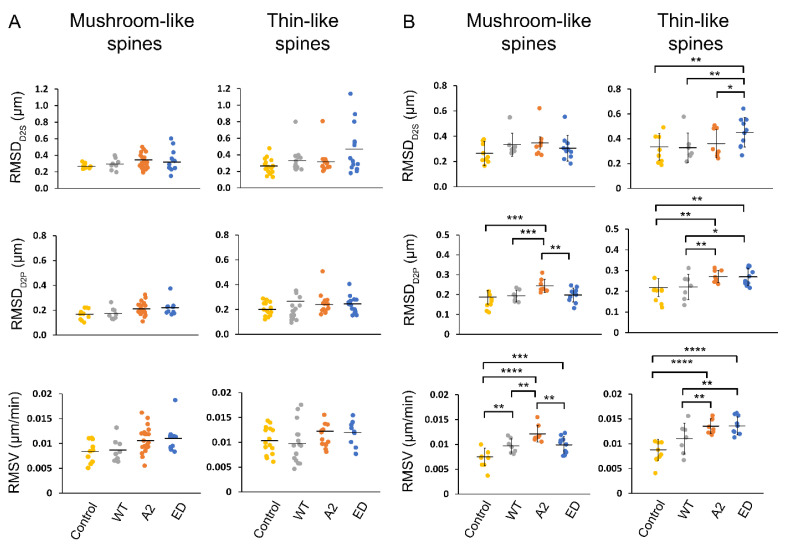
RMSD and RMSV plots for spines in an average representative dendrite (**A**) and overall dendrites measured (**B**). In (**A**), each point in the plot represents RMSD (D2S and D2P) and RMSV of each spine type in the representative dendrite over the course of 20 min time-lapse. In (**B**), each point represents the average RMSD of all spines (mushroom or thin) in a dendrite over the course of 20 min time-lapse. The mean is represented by the solid line between data points. The error bars represent the standard deviations of the plots. Asterisks indicate statistically significant differences; *, **, *** and **** represent *p*-values of *p* ≤ 0.1, *p* ≤ 0.05, *p* ≤ 0.005, and *p* ≤ 0.0005, respectively.

**Figure 7 biomolecules-13-01237-f007:**
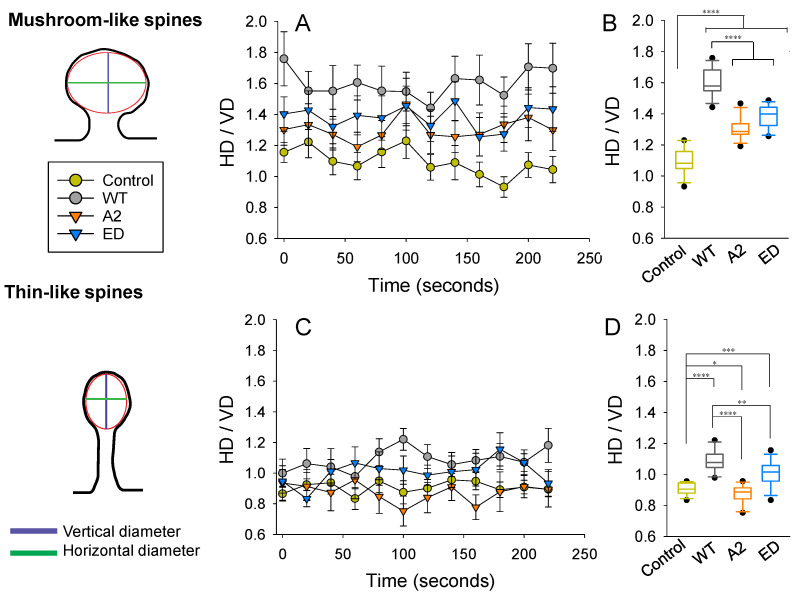
Effect of Tmod2 overexpression on spine head parameters. The ratio between horizontal (HD) and vertical (VD) diameters was calculated at each time point for mushroom (**A**,**B**) and thin (**C**,**D**) spines. (**A**,**C**) Data graphed as within-group (Control, WT Tmod2, Tmod2 A2, and Tmod2 ED) mean ± SEM (*n* = 15 for each group). (**B**,**D**) Data collapsed across time with respect to each group and graphed in box and whisker plots to illustrate distribution. Asterisks indicate differences with; *, **, ***, and **** representing *p*-values of *p* = 0.131, *p* ≤ 0.05, 0.005, and 0.0005, respectively.

**Figure 8 biomolecules-13-01237-f008:**
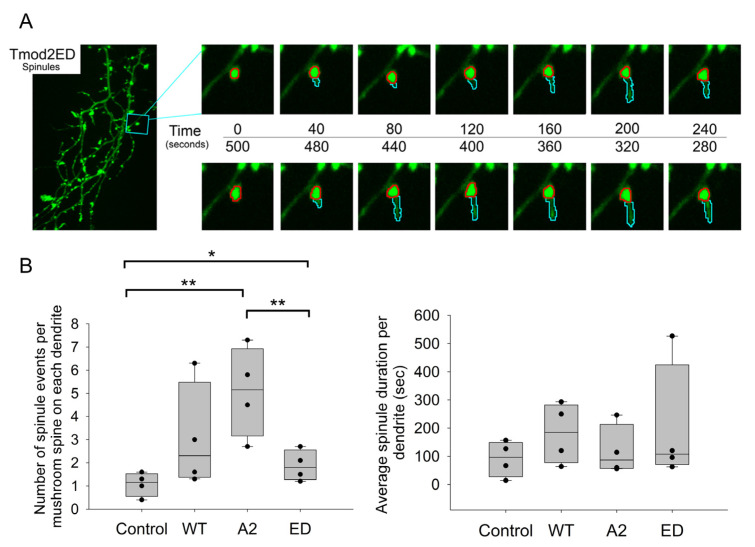
(**A**) Representative ~5 min section of a 20 min time-lapse for Tmod2 ED overexpressed neurons indicating the formation of spinules. The spine head is traced by the red circle. The formed spinule and subsequent changes in it are traced by a cyan marker. (**B**) Box plots for normalized number of spinules to total number of mushroom spines per dendrite and average spinule duration per dendrite in seconds. * and **, represent *p*-values of *p* ≤ 0.1 and *p* ≤ 0.05, respectively.

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
