# Peer review of "Effects of Tropomodulin 2 on Dendritic Spine Reorganization and Dynamics"

_biomolecules, 2023, doi:10.3390/biom13081237_

Round 1

Reviewer 1 Report

This is comprehensive spine morphology and motility analysis of primary hippocampal neurons expressing Tmod2 and its two mutations. Mutations disrupt the binding to tropomyosin or actin and they affect spines differently. Main takehome message is that “Tmod2’s nucleating and capping abilities have very specific effects on the dendritic spine formation, reorganization, and motility”, but further mechanistic details are missing.

Comments and concerns:

1. Reorganize results. Now TmodED and Tmod2A2 results are presented before telling what they are and why they are studied. Explanation comes in Figure 4. Analysis of these mutations is anyway the main point in this manuscript, so I think they can be presented in the first Figure. I suggest to explain mutations first (Figure 4) and then present results as they are presented now. Make sure that reader understands from reading results which single mutations and effects TmodED and Tmod2A2 refers.

2. Figure 1A, in addition to Tmod2WT, add panels for control, TmodED and Tmod2A2

3. Figure 3A, in addition to Tmod2WT, add panels for control, TmodED and Tmod2A2

4. Figure 5. It seems that time-lapse video moves to left. Thus, analysis is not analyzing the changes in spines but changes in imaging field. I suggest re-analyzing these videos. For example, spine dynamics can be analyzed by analyzing changes in spine head width or spine length. These analyses are not sensitive for drifting. I see from methods that StackReg plugin was implemented to correct for any observed drifting motion but based on shown figure it has not worked. Analyses can be added to the same Figure with images (now presented separately in Figures 5 and 6). Give in figure legends how many videos and how many spines are analyzed (same for spinules in Figure 7).

Reviewer 2 Report

In this paper the authors studied the effects of Tmod2 overexpression on dendritic spine morphology and dynamics in primary hippocampal neurons. Members of the Tmod protein family are presumably the most well-known F-actin pointed end binding proteins, yet their functions in regulation of the dendritic cytoskeleton is only partly understood. In this study, the effect of wild type Tmod2 was compared to two mutants forms, one of which is impaired in Tropomyosin (TM) binding (designated as ED), and another one with reduced actin binding ability (designates as A2). Spine morphology, spine number and spine length were used as major readouts in fixed samples, whereas spine motility was measured in 20 minutes videos, all obtained from transfected primary neurons. While most parameters measured showed no or non-significant changes, OE of the WT form slightly decreased spine number and increased spine length. As compared to this, the excess of ED somewhat also decreased spine number without a significant effect on length. The A2 form increased the number, as well as the length of the mushroom-like spines without significantly affecting the thin and stubby spines. Number of the dendrite shaft synapses was increased by about two-fold upon ED OE, and spinule number was increased by WT and ED OE. Regarding spine motility, the WT form had negligible effects on all parameters, while the ED, and to a lesser extent the AD forms weakly increased dynamics of the thin spines but not of the mushroom-like ones. These findings were then interpreted in the Discussion as if the actin nucleation and pointed end capping abilities of Tmod2 had specific effects on dendritic spine formation and motility. Whereas, Tmod2 was indeed reported to exhibit an actin monomer binding and a weak nucleation activity in vitro, a careful study of Tmod3 has shown that in the presence of TM these activities are suppressed, and therefore the nucleation activity is most likely irrelevant in the living cells. Given their similarities, the same is likely to hold true for Tmod2, and for this reason I have a severe concerns as to the in vivo relevance of this paper, and interpretation of the data. It is also quiet odd that a proper introduction of the mutant versions is missing, the actin nucleation versus capping activity only comes up in the Discussion.

Overall, I find that this work in the present form is not suitable for publication. The in vivo relevance (if any) of this type of overexpression analysis is highly questionable. In addition, as detailed below, numerous controls are missing, and English should be improved as well.

Specific comments:

1.         Protein expression level controls are missing. In case of OE experiments, it should be a must to know what is the OE level as compared to WT.

2.         The subcellular localization pattern of Tmod2 has already been described in dendrites with high resolution. The authors should examine the localization pattern of their overexpressed WT and mutant forms as compared to the normal situation.

3.         The authors constantly talk about actin dynamics. Surprisingly, they didn’t attempt to examine whether Tmod2 OE alters actin organization in the dendrites and spines.

4.         A pervious study reported that the knockdown of Tmod2 does not disrupt spine morphogenesis and impair synapse formation. If so, it is again questionable whether the current study has any relevance as to the in vivo situation.

5.         In lines 614-615 the authors state that “Our findings imply that Tmod2’s nucleating and capping abilities have very specific effects on the dendritic spine formation, reorganization, and motility.” This is a heavy overstatement though because it could only be concluded if rescue experiments are done in a null mutant background. In the present study we have no idea what is really measured, is it a neomorphic effect, is it an antimorphic effect, is it related to the normal cellular function of Tmod2 or is it just artifact due to a high expression level? And also, because of the presence of TM, it is much more likely that the presumed nucleation activity is not at work in this system.

I noted a number of typos and other grammatical errors, therefore correction of English usage is highly recommended. 

Reviewer 3 Report

The authors have performed a thorough microscopic analysis of the role of tropomodulin 2 (Tmod2) on spine morphology in primary hippocampal neurons. The manuscript is logically written and the findings will contribute to our understanding of the role of Tmod2 in synaptogenesis and dendritic growth. However, I would like to recommended the following to the authors.

Considering that the pointed-end of actin is also regulated by other actin-binding proteins (ABPs), as mentioned in the discussion by the authors, an additional small study that the authors should consider is to quantify how the level of overexpression of Tmod2 affects the level of expression of, e.g., Tmod1 and other ABPs as these data can provide valuable additional information. This can be easily done by Western blot or targeted mass spectrometry. 

Minor comments:

PSD should be described early in the text.

To help readers who are not familiar with neuronal proteins, could the authors provide the brief characteriszation of Homer1 and MAP2 in point 2.1.4. instead of 2.1.6.

I presume that Tables A1 and A2, will be given as supplemental data as they do not appear in the manuscript.

Round 2

Reviewer 2 Report

The revised paper is improved significantly, and the authors gave convincing answers to my questions and comments. I support the acceptance of this paper.

Minor corrections might be required.

Author Response

We thank the reviewer for taking the time to review our revised manuscript. We have proofread the manuscript once again to improve the English.

Reviewer 3 Report

While there are still some issues with the paper, I understand that due to the low number of cells being transfected, they will be difficult to address. The authors may want to consider mentioning the low yield of transfection in their results section and discussing the possible implications of the compensation of other tropomodulin isoforms.

Author Response

We thank the reviewer for taking the time to review our revised manuscript. Low transfection yield is typical for this type of cells as shown by previous publications, therefore we added a sentence about it with a corresponding citation in section 2.1.3. We also discussed upregulation of another Tmod isoform in the Conclusions.